# Assessing Gross Motor and Gait Function Using Hip–Knee Cyclograms in Ambulatory Children with Spastic Cerebral Palsy

**DOI:** 10.3390/s25144485

**Published:** 2025-07-18

**Authors:** Jehyun Yoo, Juntaek Hong, Jeuhee Lee, Yebin Cho, Taekyung Lee, Dong-wook Rha

**Affiliations:** 1Department of Rehabilitation Medicine, Gachon University Gil Medical Center, College of Medicine, Gachon University, Incheon 21565, Republic of Korea; jehyeon.yu@gmail.com; 2Department and Research Institute of Rehabilitation Medicine, College of Medicine, Yonsei University, Seoul 03722, Republic of Korea; ghdwnsxor@yuhs.ac (J.H.); cathyjhl@yonsei.ac.kr (J.L.); otyebin@yuhs.ac (Y.C.); sbzld@yuhs.ac (T.L.)

**Keywords:** cerebral palsy, gross motor function, gait function, cyclogram, digital-biomarker

## Abstract

**Highlights:**

Children with cerebral palsy (CP) often experience weakness, spasticity, and muscle shortening, leading to deficits in gross motor, gait, and selective motor functions. GMFM-66, gait analysis, and the SCALE are commonly used in this regard, but are limited by time and space constraints in hospital settings. This study aims to identify digital biomarkers to assess gross motor, gait, and selective motor functions in children with CP, using simple kinematic data for hip–knee cyclogram-based analysis.

**What are the main findings?**
The parameters of the hip–knee cyclogram showed moderately strong correlations with GMFM-66, gait speed, GDI, and the sagittal plane subscore of the GPS for the hip and knee, while the principal component ratio correlated with the SCALE.In particular, the swing phase area showed the strongest correlation and was used to build regression models to estimate the GMFM-66 and gait speed, while the principal component ratio was used to assess the SCALE.

**What are the implications of the main finding?**
Cyclogram metrics showed potential as digital biomarkers for assessing gross motor, gait, and selective motor functions in children with cerebral palsy.

**Abstract:**

Weakness, spasticity, and muscle shortening are common in children with cerebral palsy (CP), leading to deficits in gross motor, gait, and selective motor functions. While traditional assessments, such as the Gross Motor Function Measure (GMFM-66), instrumented gait analysis, and the Selective Control Assessment of the Lower Extremity (SCALE), are widely used, they are often limited by the resource-intensive nature of hospital-based evaluations. We employed cyclogram-based analysis, utilizing simple hip and knee joint kinematics to assess clinical measures, including GMFM-66, normalized gait speed, the gait deviation index (GDI), and the gait profile score (GPS). Principal component analysis was used to quantify the cyclogram shape characteristics. A total of 144 children with ambulatory spastic CP were included in the study. All the cyclogram parameters were significantly correlated with GMFM-66, gait speed, the GDI, and the sagittal plane subscore of the GPS for the hip and knee, with the swing phase area showing the strongest correlation. Regression models based on the swing phase area were used to estimate the GMFM-66 (R^2^ = 0.301) and gait speed (R^2^ = 0.484). The PC1/PC2 ratio showed a moderate correlation with selective motor control, as measured by the SCALE (R^2^ = 0.320). These findings highlight the potential of hip–knee cyclogram parameters to be used as accessible digital biomarkers for evaluating motor control and gait function in children with bilateral spastic CP. Further prospective studies using wearable sensors, such as inertial measurement units, are warranted to validate and build upon these results.

## 1. Introduction

Weakness and spasticity are common in children with cerebral palsy (CP) and often affect gross motor development, leading to gait disturbances [1]. Muscle shortening in the hip, knee, and ankle flexors also contributes to gait abnormalities in children with spastic bilateral CP [2,3]. The Gross Motor Function Measure (GMFM)-66 is a valuable tool for assessing gross motor function in children with CP. It evaluates motor skills across five dimensions, namely lying and rolling, sitting, crawling and kneeling, standing, and walking/running/jumping, providing a reliable summary score of overall gross motor ability. Gait analysis using marker-based motion capture systems can identify gait disturbances caused by bony deformities, muscle contractures, and spasticity, helping guide treatments, such as botulinum toxin injections or surgery [4,5]. Based on kinematic data obtained during gait events, the gait deviation index (GDI) quantifies gait quality by comparing 15 gait features to those of typically developing children [6]. It is widely used to assess gait in children with CP [7,8,9,10]. Similarly, the gait profile score (GPS) quantifies gait by summing the deviations between a patient’s kinematic data and normative data [11], and has been shown to represent gross motor and gait functions in individuals with CP [12,13].

However, both the GMFM-66 and instrumented gait analysis require trained professionals, expensive equipment, and ample space. Additionally, patients tend to show better performance when tested in a hospital setting compared to their gait function in real-life conditions [7,14]. Therefore, hospital-based evaluations cannot fully capture a patient’s gross motor and gait functions. To overcome these challenges, researchers are exploring digital biomarkers using simple methods, such as 2D video or wearable sensors, like inertial measurement units (IMUs) [15], in real-life settings.

The cyclogram concept, introduced by D.W. Grieve in 1968 [16], involves plotting one joint’s angle against another during the gait cycle, creating parametric curves that offer geometric insights, such as in regard to the area and perimeter [17]. Cyclograms are derived solely from kinematic joint data, which can be easily acquired using methods like 2D video or IMUs. Unlike conventional gait analysis, which focuses on individual joint movements, cyclograms enable simultaneous analysis of joint relationships and range of motion (ROM). In children with CP, muscle weakness and spasticity disrupt normal joint interactions, leading to gait patterns like equinus, crouch, and jump knee [18]. This makes cyclogram analysis useful for evaluating gait and gross motor functions in children with CP.

Several studies have demonstrated significant differences in cyclogram parameters, such as the area and perimeter, between individuals with severe and mild gait disturbances, including those with hemiplegic stroke, osteoarthritis, and multiple sclerosis [19,20,21]. Other studies have also used cyclograms to assess gait asymmetry in individuals with hemiplegic stroke [22,23]. However, these studies primarily focused on group differences and did not provide quantitative correlation analyses. To the best of our knowledge, no studies have directly assessed gross motor and gait function using cyclograms in children with CP.

Another aspect of CP is the loss of selective voluntary motor control (SVMC), which manifests as synergistic movements in the lower extremities. This adversely affects walking, reducing step length and gait speed, and lowering the gait profile score [24]. Clinically, SVMC can be evaluated using the Selective Control Assessment of the Lower Extremity (SCALE) [25]. However, the SCALE evaluates the selective movement of each joint in isolation, limiting its effectiveness in assessing dynamic movements. It also requires trained experts for accurate administration. To date, there is no method to quantitatively assess SVMC during gait, especially in an out-of-hospital setting.

We hypothesized that cyclogram-based analysis could be used to assess patients’ gross motor function, including gait and selective motor functions. Before obtaining data from out-of-hospital settings, we conducted a preliminary retrospective analysis to evaluate the correlation between hospital-based assessments and cyclogram parameters. We analyzed the quantitative relationship between hip–knee cyclogram parameters with each of the following: GMFM-66, spatiotemporal gait parameters, the GDI, and the GPS. In addition, we analyzed the correlation between the shape of the cyclogram and SVMC using principal component analysis (PCA). We also built a linear regression model to estimate the GMFM-66, gait speed, and SCALE scores, based on the cyclogram’s parameters and shape.

## 2. Materials and Methods

This study was retrospectively conducted on 144 children with bilateral spastic CP, who could walk indoors without support for at least 10 m (classified as Gross Motor Function Classification System [GMFCS] levels I, II, and III) [26], aged between 6 and 20 years (Table 1). The children were assessed using the GMFM-66, the SCALE, and instrumented gait analysis, from June 2018 to June 2023. The SCALE was used to assess selective movements of the hip, knee, ankle, subtalar, and first toe joints, with scores assigned as 2 (normal), 1 (impaired), or 0 (unable) [25]. Instrumented gait analysis was conducted using a marker-based motion capture system, using the VICON camera system (VICON MX-T10, Oxford Metrics Inc., Oxford, UK). The markers were placed on the subject using a modified Helen Hayes marker set for lower limbs, consisting of 16 reflective markers placed on the left and right anterior superior iliac spine, posterior superior iliac spine, lateral midshaft femur, lateral knee joint axis, lateral midshaft shank, lateral malleolus, calcaneus, and dorsum of the foot between the second and third metatarsal heads. Prior to data collection, the children were asked to take approximately 20 to 30 steps to allow their gait patterns to stabilize. They were instructed to walk at their most comfortable speed. Gait trials were repeated until minimal variability was observed between the gait cycles, ensuring consistency within the trial. A single representative gait cycle from the middle of the best trial was then selected to extract sagittal plane hip and knee kinematic data, using an inverse kinematics method. The gait speed was normalized according to the average leg length. The GDI and GPS were calculated using kinematic data from patients with a normal gait. This normal gait data was collected in 2003 from 15 male adults, 15 female adults, and 15 children.

The cyclograms were created using hip and knee angles from a single gait cycle, with the area defined as the space enclosed by the cyclogram (Figure 1a). The swing phase area was defined as the region between the toe-off and heel-strike points on the cyclogram (Figure 1b). The perimeter represents the total boundary length (Figure 1c). Figure 2 shows examples of cyclograms for children with GMFCS levels I, II, and III. The cyclogram parameters, including the overall area, swing phase area, and perimeter, were summed up across both legs.

We analyzed the relationships between the cyclogram parameters (overall area, swing phase area, and perimeter) and the clinical measures, including GMFM-66 scores, normalized gait speed, the GDI, and the sagittal plane hip–knee score from the GPS. The Shapiro–Wilk test was used to assess the normality of the data. The GMFM-66, normalized gait speed, GDI, GPS and cyclogram parameters met the normality assumption. Therefore, we employed Pearson’s correlation coefficients to evaluate the significance of these correlations. We built a linear regression model to estimate the GMFM-66 scores and normalized gait speed, based on the swing phase area.

To analyze the shape of the cyclogram, we employed PCA. We derived the covariance matrix of the hip and knee angles from a single gait cycle of the more involved leg, and we derived principal component 1 (PC1) and principal component 2 (PC2) (Figure 1d). Then we calculated the ratio of PC1 to PC2, an index used to evaluate SVMC, and compared it to the SCALE score of the more involved leg. The SCALE and PC1/PC2 did not meet the normality assumption in the Shapiro–Wilk test; therefore, we used the Spearman rank correlation coefficient. Additionally, we built a linear regression model to estimate the SCALE, using the ratio of PC1 to PC2 (Figure 3).

All the analyses were conducted using SPSS (version 27.0, IBM, Chicago, IL, USA), and significance was considered to occur at *p* < 0.05. Correlation coefficients ≤ 0.3 were considered poor or weak correlations, 0.3 to 0.5 as fair, 0.5 to 0.8 as moderately strong, and ≥0.8 as very strong [27]. This study was approved by the Institutional Review Board at Yonsei University (IRB approval number: 4-2024-0044).

## 3. Results

The correlations between the cyclogram parameters and the clinical parameters are shown in Table 2. All the parameters showed a correlation with GMFM-66, the normalized gait speed, the GDI, and the sagittal plane of the hip and knee subscore of the GPS. The swing phase area showed a moderately strong positive correlation with GMFM-66 (R = 0.55, *p* < 0.05) and the normalized gait speed (R = 0.70, *p* < 0.05), and a fairly positive correlation with the GDI (R = 0.41, *p* < 0.05) and the sagittal plane of the hip and knee subscore of the GPS (R = 0.48, *p* < 0.05). The linear regression model for estimating the GMFM-66 and normalized gait speed using the swing phase area is shown in Table 3 (R^2^ = 0.301 for GMFM-66 and R^2^ = 0.484 for gait speed).

The results of the subgroup analysis are presented in Table 4. In the multiple comparison test, using the two-sample *t*-test, the cyclogram parameters showed statistically significant differences across the GMFCS groups. In the correlation analysis, the GMFCS level II group displayed correlations similar to those of the overall patient group, whereas the GMFCS level I group exhibited weaker correlations. For the GMFCS level III group, only the swing phase showed a correlation with GMFM-66 and gait speed.

There was a moderately strong negative correlation between the eigenvalue ratio of the swing phase and the SCALE in Table 2 (ρ = −0.57, *p* < 0.05). The linear regression model for estimating the SCALE using the eigenvalue ratio of the swing phase is shown in Table 3 (R^2^ = 0.320). Scatter plots of the cyclogram parameters and clinical parameters are provided in the Appendix A.

## 4. Discussion

In this study, we found that specific hip–knee cyclogram parameters were significantly correlated with gross motor function, as measured by the GMFM-66, in ambulatory children with CP (GMFCS levels I–III). These parameters also showed meaningful associations with other gait-related measures, including gait speed, the GDI, and the GPS. Moreover, we developed a linear regression model to estimate the GMFM-66 scores using the swing phase area, highlighting the potential of cyclogram-based metrics to be used as predictive markers of gross motor function. In addition, we found that the variable quantifying the shape of the cyclogram using PCA was significantly correlated with the SCALE score, which assesses selective motor control in patients. A previous study found that spatiotemporal parameters were correlated with gross motor function [28]. However, spatiotemporal parameters, including gait speed, primarily reflect locomotor ability, and do not capture gait quality, which requires consideration of hip and knee kinematics. In contrast, cyclogram-based analysis can provide detailed information on hip and knee kinematics and functional ROM during gait events. Gait disturbances in patients with spastic bilateral CP are often caused by shortened and spastic lower extremity muscles, leading to decreased ROM in the hip and knee joints. Previous studies have shown that children with CP typically have decreased joint ROM compared to children without CP [29], and spasticity and weakness of the hip muscles cause a reduction in ROM of the hip joint [30]. Similarly, spasticity in the knee joint muscles reduces the angular velocity of both knee flexion and extension, further limiting knee ROM [31]. Several studies have demonstrated a correlation between the degree of limited ROM on physical examination and the severity of gross motor function in children with CP [29,32]. This diminished ROM in the hip and knee joints contributes to the reduced area and perimeter of the cyclogram observed in patients with CP. Therefore, it may serve as a useful indicator for identifying reductions in functional range of motion during gait events caused by spasticity or muscle shortening, and for determining the need for therapeutic interventions, such as botulinum toxin injections or surgical procedures.

Generally, as the gait speed increases, the ROM of the hip and knee joints also increases [31]. Therefore, it seems natural that the cyclogram parameters would increase as the gait speed increases. However, during the gait analysis, we instructed the children to walk at a comfortable pace. A study found that comfortable gait speed is a useful indicator of gait function in individuals with neurological disorders [33]. This suggests that the normalized gait speed could represent the overall gait function, and the cyclogram could serve as a useful metric for assessing gait function.

We also found a correlation between the cyclogram parameters with the GDI and the sagittal plane subscores of the hip and knee joints from the GPS. The GDI and GPS focus on deviations from normal gait, and, as previously mentioned, a decreased ROM can lead to reduced cyclogram parameters, resulting in greater deviations from typical gait patterns. Previous studies have shown a correlation between the degree of gait deviation and the weakness, spasticity, and joint contracture of children with CP [34,35]. Both the GDI and GPS showed moderate correlations, with the GPS demonstrating a slightly stronger association than the GDI. While the GDI is derived from various gait kinematic features in three-dimensional space, the GPS in this study was limited to the sagittal plane of the hip and knee, which may explain the stronger correlation. The fact that the cyclogram parameters were correlated with both the GDI and GPS suggests their potential utility in assessing gait quality, not only reflecting the gait speed, but also the degree of deviation from normal gait kinematics. This implies that cyclogram parameters could support clinical decision-making when aiming to normalize gait patterns through interventions such as botulinum toxin injections or surgical procedures. Furthermore, they may be useful in evaluating treatment effects by comparing gait patterns before and after an intervention.

Although the GDI and GPS showed statistically significant correlations, their significance was lower compared to GMFM-66 and the normalized gait speed. This may suggest that achieving a high GDI score may not be an appropriate goal for children with spastic CP, who have impairments, such as bone deformities, muscle weakness, and spasticity [36]. These pathological conditions result in reduced gait efficiency compared to typically developing children [37]. In such cases, aiming for normal gait kinematics may not be an achievable or appropriate goal for children with CP. Some studies have shown that while the GDI improved after surgical treatment, the gross motor function, gait speed, and energy efficiency did not improve in all patients [10,38]. On the other hand, the cyclogram parameters represent the ROM during gait events, which can improve through muscle strengthening or cardiopulmonary fitness, not just surgery. However, such improvements may not always bring joint kinematics closer to the normal range.

Among the hip–knee cyclogram parameters, the swing phase area showed the strongest correlation with gross motor function and gait speed compared to other parameters. Impairments in motor coordination of the hip and knee joints are common in spastic bilateral CP, and these difficulties are more pronounced during the swing phase compared to the stance phase [39]. During the swing phase, spasticity and shortening of the medial hamstring muscles restrict full knee extension. A previous study showed that children with spastic CP who have poor SVMC exhibit limited knee extension during the swing phase [40]. Another study found that children with CP who have spastic hamstrings and quadriceps show slower knee angular velocities during the swing phase [31]. However, during the stance phase, ground reaction forces can influence joint ROM. For example, equinus gait caused by spastic paralysis results in genu recurvatum, due to the coupling between ankle plantar flexors and knee extensors, which is influenced by the direction of ground reaction forces [41,42]. This suggests that the swing phase area may more directly reflect gross motor function and gait speed than the overall area of the cyclogram.

The perimeter showed a weaker correlation with the clinical parameters than the overall area and swing phase area. In cases where patients have good gross motor function, an increased ROM leads to an increase in the perimeter. However, if children lose the ability to control joint movements smoothly within the same ROM, the cyclogram path exhibits irregular and more extensive excursions, increasing the perimeter despite a reduction in area. This could explain the weak correlation of the perimeter with the clinical parameters.

In the subgroup analysis, there is a statistically significant difference between the GMFCS levels. In the correlation analysis for the GMFCS level I group, the cyclogram parameters showed weaker correlations with GMFM-66 and the normalized gait speed, likely due to a ceiling effect, as most children could walk independently and had near-perfect GMFM-66 scores. In the GMFCS level III group, a statistically non-significant weak correlation was observed between the cyclogram parameters and the clinical values. GMFCS level III is defined as children who can walk indoors with assistive devices. Notably, some children in this group can walk up to 10 m indoors without assistance, and we included these children in this study. However, these children tend to show greater variability in their abilities compared to those in the GMFCS level I or II groups. The limited sample size of 17 GMFCS level III children may have contributed to this weak correlation.

Although CP often affects the distal ankle joint more than the hip or knee, this study focused on the hip and knee joints. This decision was made because hip–ankle and knee–ankle cyclograms have not been validated in previous studies, whereas parameters such as the area and perimeter have been examined for hip–knee cyclograms [19,20,21]. Moreover, hip–ankle and knee–ankle cyclograms typically form a figure-eight shape, making area and perimeter calculations more complex. Additionally, we plan to conduct a prospective study using wearable sensors, such as IMUs, and attaching these sensors to the distal ankle joint is more challenging compared to the larger hip and knee joints.

SVMC is also an essential factor, despite its limited exploration in relation to gross motor function [43]. The SCALE score, which evaluates SVMC, has been shown to correlate with impaired gait ability [24]. This is because spasticity, weakness, synergistic movements, and incoordination, often resulting from corticospinal tract damage, can reduce movement control complexity [44]. These non-selective movements are more pronounced in children with CP, who have more significant functional limitations, as assessed by the GMFCS levels [45]. The hip and knee joints exhibit pathological, non-dissociative movement, where the knee cannot fully extend, while the hip flexes during the swing phase, due to spasticity and shortening of the medial hamstring muscles. This non-dissociative movement causes the hip and knee joints to flex simultaneously during mid-swing and extend during a terminal stance. A previous study found that the loss of SVMC disrupts the relative phase between the hip and knee joints, as the knee cannot fully extend during hip flexion at terminal swing, leading to an elongation of the cyclogram’s shape [39]. Using PCA, we quantified the elongation of the hip–knee cyclogram by calculating the ratio of the two principal components (PC1 to PC2) from the hip and knee angle data. The eigenvalue ratio correlated with the SCALE score, indicating a link to SVMC and the potential to estimate the SCALE using a linear regression model.

When assessing gross motor function and gait speed in children with bilateral spastic CP, the cyclogram parameters for both legs were averaged since both legs are critical to these assessments. However, for the SVMC assessment, summing up or averaging the function of both legs would not provide an accurate representation. Although all participants had bilateral CP, each child typically had one side that was more affected than the other. Therefore, we focused on analyzing the more affected side, as indicated by the SCALE score.

Conventional assessments of gross motor function and SVMC in children with CP require hospital visits and costly equipment, like instrumented gait analysis, which may not fully capture a child’s abilities in daily environments. In contrast, cyclogram parameters derived from simple kinematic data, such as hip and knee angles, offer a cost-effective alternative. These data can be collected using wearable sensors like IMUs [46] or through 2D video analysis, involving keypoint detection algorithms. This approach enables motor function assessment outside clinical settings, providing real-world insights into children’s abilities during everyday activities.

## 5. Study Limitations

This study has several limitations. First, it was a retrospective study, and most patients underwent gait analysis to evaluate their gait patterns prior to orthopedic surgery for bone deformities or muscle spasticity/shortening. As a result, a large proportion of the participants were classified as GMFCS level II, which may have contributed to selection bias. Another important limitation of this study is that the analysis was based on a single time-point assessment of the cyclogram and its correlation with gross motor and gait functions. Therefore, it remains unclear whether cyclogram parameters are sensitive to functional changes following therapeutic interventions or clinical events. Further longitudinal studies are needed to determine whether cyclograms can effectively capture dynamic changes in gait and motor function in response to interventions or disease progression.

Although the study focused on children with bilateral spastic CP, the degree of involvement between the two limbs can vary. However, when calculating cyclogram parameters, such as the area, swing phase area, and perimeter, we used the average values of both limbs. If it had been possible to quantify the more involved side and apply greater weight to it during parameter calculation, a more realistic assessment could have been achieved. The retrospective nature of the study limited our ability to reflect the functional variation between limbs.

Lastly, while this study highlighted that the main advantage of cyclograms lies in their reliance solely on kinematic data, which does not require hospital-based instrumented gait analysis, we used data obtained through instrumented gait analysis in a laboratory setting. Nevertheless, this study was conducted to explore the potential of cyclograms as digital biomarkers using simple kinematic data, as a preliminary step before conducting prospective studies. Future prospective studies using wearable sensors are warranted to validate these findings in real-world environments.

## 6. Conclusions

Hip–knee cyclogram parameters, particularly the swing phase area, which can be easily obtained from simple kinematic data of the sagittal plane, could serve as potential digital biomarkers for assessing gross motor and gait functions in children with bilateral spastic CP. The PC1/PC2 of the cyclogram, derived through the use of PCA, could also be a useful metric for assessing SVMC during gait events, providing an alternative to traditional hospital-based evaluations.

## Figures and Tables

**Figure 1 sensors-25-04485-f001:**
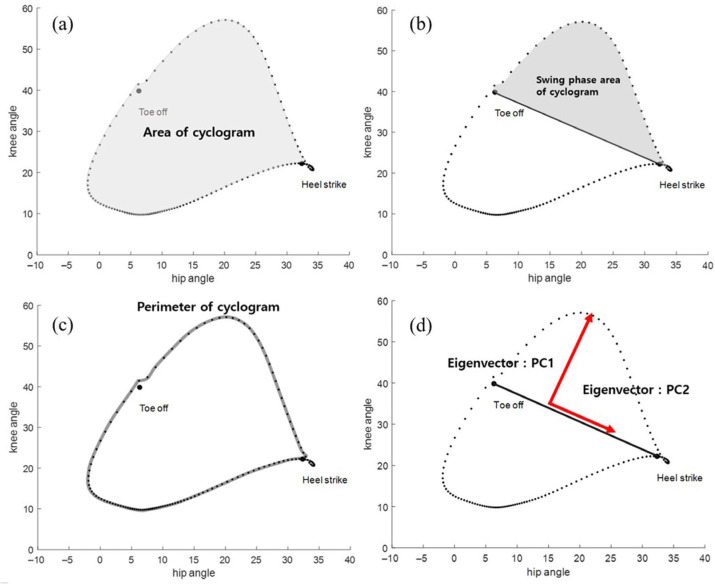
Definition of parameters of cyclograms. (**a**) Cyclogram parameter: area. (**b**) Cyclogram parameter: swing phase area. (**c**) Cyclogram parameter: perimeter. (**d**) Cyclogram parameter: PC1/PC2 (PC1: principal component 1, PC2: principal component 2).

**Figure 2 sensors-25-04485-f002:**
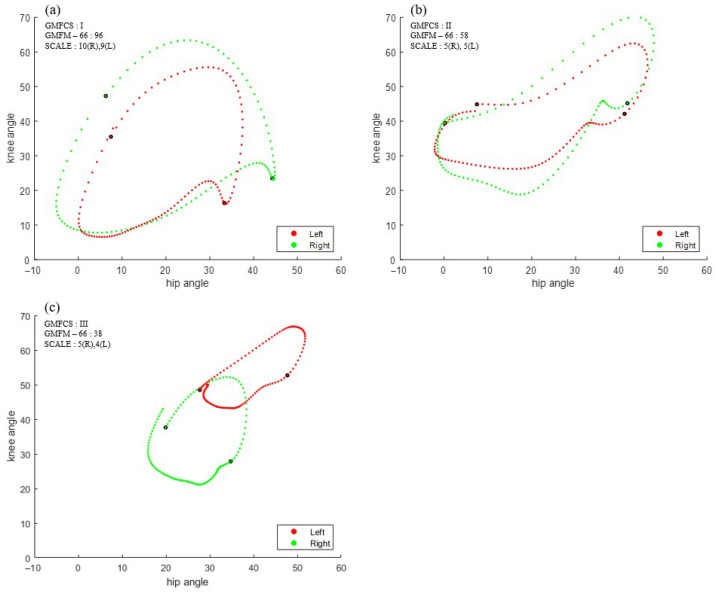
Examples of cyclograms from children with CP: (**a**) GMFCS I; (**b**) GMFCS II; and (**c**) GMFCS III (GMFCS: Gross Motor Function Classification System).

**Figure 3 sensors-25-04485-f003:**
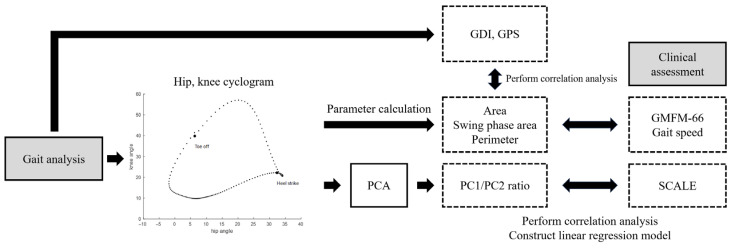
Overview of the study workflow (GMFM-66: Gross Motor Function Measure-66, PCA: principal component analysis, PC1: principal component 1, PC2: principal component 2, SCALE: Selective Control Assessment of the Lower Extremity, GDI: gait deviation index; GPS: gait profile score. Gait speed was normalized according to the average leg length.

**Table 1 sensors-25-04485-t001:** Demographic information (*n* = 144).

Demographic Variable	No. (%)
Sex	
Male	82 (56.94%)
Female	62 (43.06%)
GMFCS	
I	36 (25.00%)
II	91 (63.19%)
III	17 (11.80%)
Age (years)	
Range	6–20
Average	13 years
Mean	13 years 8 months
Standard deviation	3 years 6 months

GMFCS: Gross Motor Function Classification System.

**Table 2 sensors-25-04485-t002:** Correlation between cyclogram parameters (area, swing phase area, perimeter) and GMFM-66, normalized gait speed, GDI, and correlation between PC1/PC2 and SCALE.

	Area	Swing Phase Area	Perimeter	PC1/PC2
GMFM-66	0.48 *	0.55 *	0.41 *	
Normalized gait speed	0.60 *	0.70 *	0.66 *	
GDI	0.48 *	0.41 *	0.34 *	
GPS (hip fl/ex, knee fl/ex)	−0.51 *	−0.48 *	−0.41 *	
SCALE				−0.57 *

GMFM-66: Gross Motor Function Measure-66, PC1: principal component 1, PC2: principal component 2, SCALE: Selective Control Assessment of the Lower Extremity, GDI: gait deviation index, GPS: gait profile score, fl: flexion, ex: extension, * *p* < 0.05.

**Table 3 sensors-25-04485-t003:** Univariate linear regression analysis between swing phase area and GMFM-66, normalized gait speed, SCALE, and PC1/PC2.

GMFM-66	B*	β (95% CI)	*p*	R^2^
Constant	54.779			0.301
Swing phase area	0.010	0.549	<0.001	
Normalized gait speed	B*	β (95% CI)	*p*	R^2^
Constant	0.672			0.484
Swing phase area	0.0004	0.695	<0.001	
SCALE	B*	β (95% CI)	*p*	R^2^
Constant	6.413			0.320
PC1/PC2	−0.211	−0.566	<0.001	

GMFM-66: Gross Motor Function Measure-66, PC1: principal component 1, PC2: principal component 2, SCALE: Selective Control Assessment of the Lower Extremity. B*: Standardized Coefficient of linear regression model.

**Table 4 sensors-25-04485-t004:** (**a**) Mean and standard deviation of cyclogram parameters by subgroup and comparative analysis across GMFCS levels using two-sample *t*-tests. (**b**) Pearson correlation of cyclogram parameters with GMFM-66 and comfortable normalized gait speed in the subgroup analysis, based on GMFCS levels.

(**a**)						
	GMFCS I (n = 36)	GMFCS II (n = 91)	GMFCS III (n = 17)	Multiple comparisons		
				I vs. II	I vs. III	II vs. III
Area	1270.7 ± 334.4	1002.6 ± 409.4	557.9 ± 174.1	<0.001	<0.001	<0.001
Swing phase area	762.5 ± 247.9	530.2 ± 282.5	238.7 ± 115.8	<0.001	<0.001	<0.001
Perimeter	158.5 ± 18.4	142.9 ± 29.3	118.4 ± 21.8	<0.001	<0.001	<0.001
(**b**)						
	GMFCS I		GMFCS II		GMFCS III	
	GMFM-66	Gait speed	GMFM-66	Gait speed	GMFM-66	Gait speed
Area	0.20	0.49 *	0.33 *	0.56 *	0.38	0.01
Swing phase area	0.29	0.56 *	0.39 *	0.67 *	0.49 *	0.61 *
Perimeter	0.30	0.60 *	0.23 *	0.63 *	0.42	0.37

GMFCS: Gross Motor Function Classification System, GMFM-66: Gross Motor Function Measure-66, * *p* < 0.05; gait speed was normalized according to the average leg length.

## Data Availability

The datasets presented in this article are not readily available because of hospital policy restricting the external sharing of patient data.

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
