# Peer review of "Assessing Gross Motor and Gait Function Using Hip–Knee Cyclograms in Ambulatory Children with Spastic Cerebral Palsy"

_sensors, 2025, doi:10.3390/s25144485_

Round 1
Reviewer 1 Report
Comments and Suggestions for Authors
Review of "Assessing Gross Motor and Gait Function Using Hip-Knee Cyclogram in Ambulatory Children with Spastic Cerebral Palsy"
The authors have done excellent work assessing gait function using hip-knee cyclograms in ambulatory children with spastic cerebral palsy. There is a lack of published literature evaluating and interpreting these graphs, making this study particularly valuable. It's worth noting that some companies have now incorporated these cyclograms into clinical gait reports (for example: report.qualisys.com/r/f8pMFlrTA/details).
There are a few minor recommendations and suggestions for future studies.
Point 1. Normative data update needed
The authors used normative data collected in 2003, which is now over 20 years old. Future research would benefit from collecting updated normative data with more detailed categories:
• Different age groups
• Various walking speeds (slow, self-selected, average and fast)
• Both barefoot and shoe conditions
Walking speed significantly changes kinematics, kinetics, and EMG data, as do age and footwear. More advanced normative data would allow better interpretation of findings and stronger conclusions.
Point 2. There are no any single limitation in the article as a suggestion could be:
motion capture technology in 2003 had several limitations compared to current systems. The biomechanical models’ limitations as well as camera specifications had disadvantages over current systems that may affect data quality and interpretation.
Point 3. Minor formatting
It is recommended to format your text to evenly distribute across the margins for better presentation (Ctrl+J in Word).
Overall, this is valuable work on an important topic. The authors have contributed to understanding gait patterns in children with CP using cyclorams. Interestingly, I've noticed several organizations have recently shown increased interest in hip-knee flexion cyclograms, making this publication particularly relevant.
Author Response
We sincerely appreciate the reviewers’ comments that proved valuable in improving the quality of our paper. We have addressed all the comments and revised our manuscript accordingly. Our responses to all the comments and the major revisions are listed below.
The authors have done excellent work assessing gait function using hip-knee cyclograms in ambulatory children with spastic cerebral palsy. There is a lack of published literature evaluating and interpreting these graphs, making this study particularly valuable. It's worth noting that some companies have now incorporated these cyclograms into clinical gait reports (for example: report.qualisys.com/r/f8pMFlrTA/details).
There are a few minor recommendations and suggestions for future studies.
Point 1. Normative data update needed
The authors used normative data collected in 2003, which is now over 20 years old. Future research would benefit from collecting updated normative data with more detailed categories:
• Different age groups
• Various walking speeds (slow, self-selected, average and fast)
• Both barefoot and shoe conditions
Walking speed significantly changes kinematics, kinetics, and EMG data, as do age and footwear. More advanced normative data would allow better interpretation of findings and stronger conclusions.
[Response] Thank you for your valuable comment. We agree that having normative data under various conditions would have enabled a more comprehensive analysis and discussion. However, in this study, we compared cyclogram parameters only to GDI and the sagittal plane GPS scores, which reflect normative kinematic patterns and do not account for variations in walking speed or footwear conditions. This represents a limitation of our study.
Point 2. There are no any single limitation in the article as a suggestion could be:
motion capture technology in 2003 had several limitations compared to current systems. The biomechanical models’ limitations as well as camera specifications had disadvantages over current systems that may affect data quality and interpretation.
[Response] Thank you for your comments. We have added the Study Limitations section (Lines 340-364).
Point 3. Minor formatting
It is recommended to format your text to evenly distribute across the margins for better presentation (Ctrl+J in Word)
[Response] Thank you for your suggestion. We have reformatted the entire manuscript accordingly.
Overall, this is valuable work on an important topic. The authors have contributed to understanding gait patterns in children with CP using cyclorams. Interestingly, I've noticed several organizations have recently shown increased interest in hip-knee flexion cyclograms, making this publication particularly relevant.
[Response] Thank you for your comments. Based on this pilot study, we plan to conduct a prospective study using wearable sensors in real-world settings.

Reviewer 2 Report
Comments and Suggestions for Authors
Thank you for giving me the chance to review this manuscript. I found the topic interesting, and the study addresses an important clinical need with a novel and promising approach. Overall, the manuscript is generally well-structured, with results clearly presented and supported by an adequate sample size. I believe it can make a valuable contribution to the field after addressing a few clarifications and making some text revisions. Please find my section-by-section comments below.
Abstract
The abstract provides a clear description of the background, methods, and key results, but it lacks a concluding statement summarizing the clinical relevance or potential implications of the findings. I recommend adding 1–2 sentences at the end of the abstract to highlight the significance of the study and suggest future directions.
Introduction
- Line 49: You mention the (GMFM)-66 as “a valuable tool for assessing gross motor function in children with CP” but no further description of what this tool measures or how it works. Please briefly describe the GMFM-66 (e.g. items included and scoring)
- Lines 67-81: Cyclograms have been used in previous research not only to assess joint ROM but also inter-limb symmetry in conditions such as stroke (Marrone et al., 2022, https://doi.org/10.3390/sym14081560; Pilkar et al., 2018, doi:10.1115/1.4040774). Adding one or two sentences to highlight these boreader applications would help readers understand why cyclograms can be a powerful tool for analyzing gait patterns.
Methods
- Line 106-114: You mention the “plug-in gait model” (line 114 but provide no refences or details about markers placement. Please, add more information about the set-up for gait analysis, including a brief description of the marker positions, or a figure to illustrate them
- Line 136-151: This section describes several consecutive analysis steps (normality tests, correlation analyses, PCA, regression modeling) that are quite I suggest adding a simple workflow diagram summarizing the data processing and analysis pipeline.
- line 119: in the caption, “system” instead of “sytem”
- line 139: “principal” instead of “principle”
- line 148: please use “… an index to evaluate SVMC” rather than “… serves as an evaluation index for SVMC”
Results
- In the caption of table 2 and 3: “principal” instead of “principle”
- Table 4 (below line 190): please check and adjust the table formatting to ensure that the labels and values are clearly displayed and easy to read.
- Table 4: I would suggest specifying in the headings the sample size of each subgroup, since according to Table 1 the distribution appears unbalanced (e.g., n = 91 in GMFCS II vs. n < 20 in the other groups). Do you believe that this difference might have affected the results?
Discussion
- General comment: This section feels somewhat long and, at times, repetitive. I suggest revising the Discussion and shortening it by summarizing the key points more concisely to improve overall clarity. Additionally, I recommend more clearly highlighting the study’s limitations; although some are mentioned throughout the section, they are not explicitly acknowledged as limitations (for example, the imbalance in subgroup sizes). In addition, I suggest opening the Discussion with a brief summary of the main findings before moving into the interpretation. This would help orient readers and clearly link the Discussion to the Results section.
- Lines 272-291: This part is unclear. In your study you used a marker-based system to measure joint kinematics, but here you discuss the limitations of IMUs for measuring ankle motion. Please clarify the meaning of this paragraph. Since marker-based systems can reliably capture ankle and knee kinematics, it would be helpful to explain more clearly why knee–ankle cyclograms were not considered.
Author Response
We sincerely appreciate the reviewers’ comments that proved valuable in improving the quality of our paper. We have addressed all the comments and revised our manuscript accordingly. Our responses to all the comments and the major revisions are listed below.
Abstract
Commet 1 : The abstract provides a clear description of the background, methods, and key results, but it lacks a concluding statement summarizing the clinical relevance or potential implications of the findings. I recommend adding 1–2 sentences at the end of the abstract to highlight the significance of the study and suggest future directions.
[Response] Thank you for your comment. We have added the key findings of this study at the end of the abstract (Lines 41–45).
Introduction
Commet 2 : Line 49: You mention the (GMFM)-66 as “a valuable tool for assessing gross motor function in children with CP” but no further description of what this tool measures or how it works. Please briefly describe the GMFM-66 (e.g. items included and scoring)
[Response] Thank you for your comment. We have added an explanation of the GMFM-66 and its subdimensions (Lines 54–56).
Commet 3 : Lines 67-81: Cyclograms have been used in previous research not only to assess joint ROM but also inter-limb symmetry in conditions such as stroke (Marrone et al., 2022, https://doi.org/10.3390/sym14081560; Pilkar et al., 2018, doi:10.1115/1.4040774). Adding one or two sentences to highlight these boreader applications would help readers understand why cyclograms can be a powerful tool for analyzing gait patterns.
[Response] Thank you for your suggestion. We have cited papers that utilize cyclograms, as you mentioned (Lines 83–85).
Methods
Commet 4 : Line 106-114: You mention the “plug-in gait model” (line 114 but provide no refences or details about markers placement. Please, add more information about the set-up for gait analysis, including a brief description of the marker positions, or a figure to illustrate them
[Response] Thank you for your suggestion. We have provided more details about the gait analysis, including the marker set used in the study (Line 112-124)
Commet 5 : Line 136-151: This section describes several consecutive analysis steps (normality tests, correlation analyses, PCA, regression modeling) that are quite I suggest adding a simple workflow diagram summarizing the data processing and analysis pipeline.
[Response] Thank you for your suggestion. We have added a diagram illustrating the study workflow (Figure 3).
Commet 6 : line 119: in the caption, “system” instead of “sytem”
[Response] Thank you for pointing out the typographical errors. We have corrected the typo.
Commet 7 : line 139: “principal” instead of “principle”
[Response] Thank you for pointing out the typographical errors. We have corrected them throughout the entire manuscript.
Commet 8 : line 148: please use “… an index to evaluate SVMC” rather than “… serves as an evaluation index for SVMC”
[Response] Thank you for your suggestion. We revised the sentence to make it clearer. (Line 156-157)
Results
Commet 9 : In the caption of table 2 and 3: “principal” instead of “principle”
[Response] Thank you for pointing out the typographical errors. We have corrected them throughout the entire manuscript.
Commet 10 : Table 4 (below line 190): please check and adjust the table formatting to ensure that the labels and values are clearly displayed and easy to read.
[Response] Thank you for your suggestion. We have adjusted the table formatting to improve readability.
Commet 11 : Table 4: I would suggest specifying in the headings the sample size of each subgroup, since according to Table 1 the distribution appears unbalanced (e.g., n = 91 in GMFCS II vs. n < 20 in the other groups). Do you believe that this difference might have affected the results?
[Response] Thank you for your comment. This is a retrospective study, and most patients underwent gait analysis to evaluate gait patterns prior to orthopedic surgery. Therefore, a large proportion of the patients belonged to GMFCS level II. We have added this point to the study limitations section (Lines 340–350).
Discussion
Commet 12 : General comment: This section feels somewhat long and, at times, repetitive. I suggest revising the Discussion and shortening it by summarizing the key points more concisely to improve overall clarity. Additionally, I recommend more clearly highlighting the study’s limitations; although some are mentioned throughout the section, they are not explicitly acknowledged as limitations (for example, the imbalance in subgroup sizes). In addition, I suggest opening the Discussion with a brief summary of the main findings before moving into the interpretation. This would help orient readers and clearly link the Discussion to the Results section.
[Response] Thank you for your comments. We revised the manuscript according to your suggestion by opening the Discussion section with a brief summary of the main findings. We also made the Discussion section more concise. (Line 208-216)
Commet 13 : Lines 272-291: This part is unclear. In your study you used a marker-based system to measure joint kinematics, but here you discuss the limitations of IMUs for measuring ankle motion. Please clarify the meaning of this paragraph. Since marker-based systems can reliably capture ankle and knee kinematics, it would be helpful to explain more clearly why knee–ankle cyclograms were not considered.
[Response] Thank you for your comments. The primary reason we used the hip–knee cyclogram rather than involving the ankle joint is that few previous studies have utilized hip–ankle or knee–ankle cyclograms, likely due to the complex and variable movements of the ankle during gait. Additionally, this was a preliminary study aimed at exploring the potential of cyclograms as digital biomarkers using simple kinematic data, prior to conducting prospective studies with wearable sensor setups. Although ankle joint movement can be accurately captured using a marker-based motion capture system, future studies using wearable sensors must consider factors such as usability and wearability. Therefore, we selected the hip and knee joints for analysis. We have added this explanation to the study limitations section (Line 358-364).
